# Exploring the Common Appearance-Boundary Adaptation for Nighttime Optical Flow

**Hanyu Zhou**[1], **Yi Chang**[1]*, **Haoyue Liu**[1], **Wending Yan**[2], **Yuxing Duan**[1], **Zhiwei Shi**[1], **Luxin Yan**[1]

[1]National Key Lab of Multispectral Information Intelligent Processing Technology, School of Artificial Intelligence and Automation, Huazhong University of Science and Technology
[2]Huawei International Co. Ltd.
`{hyzhou,yichang,yanluxin}@hust.edu.cn`

## ABSTRACT

We investigate a challenging task of nighttime optical flow, which suffers from weakened texture and amplified noise. These degradations weaken discriminative visual features, thus causing invalid motion feature matching. Typically, existing methods employ domain adaptation to transfer knowledge from auxiliary domain to nighttime domain in either input visual space or output motion space. However, this direct adaptation is ineffective, since there exists a large domain gap due to the intrinsic heterogeneous nature of the feature representations between auxiliary and nighttime domains. To overcome this issue, we explore a common-latent space as the intermediate bridge to reinforce the feature alignment between auxiliary and nighttime domains. In this work, we exploit two auxiliary daytime and event domains, and propose a novel common appearance-boundary adaptation framework for nighttime optical flow. In appearance adaptation, we employ the intrinsic image decomposition to embed the auxiliary daytime image and the nighttime image into a reflectance-aligned common space. We discover that motion distributions of the two reflectance maps are very similar, benefiting us to *consistently* transfer motion appearance knowledge from daytime to nighttime domain. In boundary adaptation, we theoretically derive the motion correlation formula between nighttime image and accumulated events within a spatiotemporal gradient-aligned common space. We figure out that the correlation of the two spatiotemporal gradient maps shares significant discrepancy, benefitting us to *contrastively* transfer boundary knowledge from event to nighttime domain. Moreover, appearance adaptation and boundary adaptation are complementary to each other, since they could jointly transfer global motion and local boundary knowledge to the nighttime domain. Extensive experiments have been performed to verify the superiority of the proposed method.

## 1 INTRODUCTION

Optical flow is to model the dense correspondence between adjacent frames. Existing optical flow methods (Sun et al., 2018; Teed & Deng, 2020) mainly focus on the natural clean scenes while the practical yet challenging nighttime optical flow has been less investigated. The main difficulty lies in the negative influence of the nighttime degradations such as weakened texture and amplified noise. The degradation factors have unexpectedly violated the brightness constancy assumptions, which greatly weaken the discriminative visual features, and thus cause invalid motion features matching.

An intuitive solution is to resort to an auxiliary domain as the source domain, and transfer knowledge from the source domain to the target nighttime domain. The existing nighttime optical flow methods directly transfer the knowledge in either input visual space or output motion space. For example, in Fig. 1 (a) (visual space adaptation), Zheng et al. (2020) transformed visual features of source daytime domain to target nighttime domain via noise model in visual space. In Fig. 1 (b) (motion space adaptation), Li et al. (2021) used gyroscope data as the source domain to model the background motion, which assistantly improved motion features of target nighttime domain in motion space. However, these direct adaptation methods would easily suffer from distribution misalignment issue,

---

*Corresponding author.

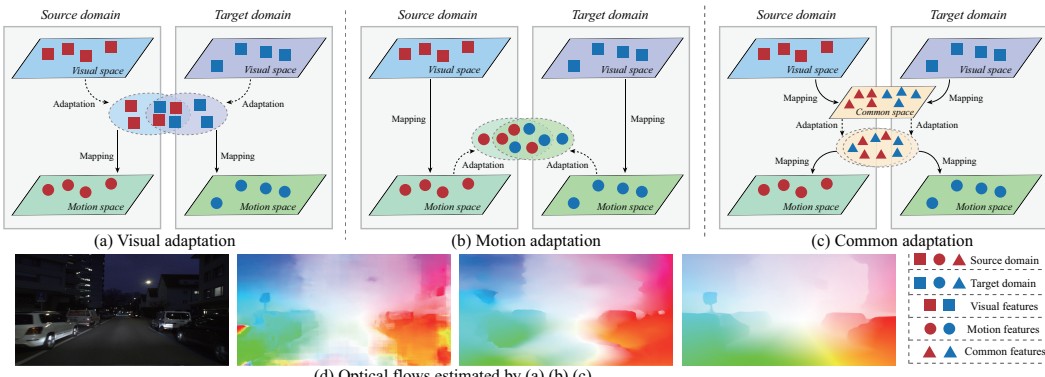

Figure 1: Illustration of three nighttime optical flow paradigms. Visual adaptation and motion adaptation directly transfer knowledge from source to target domain in input visual space and output motion space, respectively. However, this direct adaptation is ineffective due to the large domain gap caused by the intrinsic heterogeneous nature (feature distribution misalignment) of the feature representations between source and target domains. In contrast, we explore a common-latent space as the intermediate bridge to reinforce the feature alignment between the two domains. In this work, we employ daytime and event domains as the source domains, and build the reflectance-aligned and spatiotemporal gradient-aligned common spaces to transfer knowledge to target nighttime domain.

due to the intrinsic heterogeneous nature (distribution misalignment caused by degradations) of feature representations between source and target domains. *Therefore, selecting an appropriate auxiliary domain and embedding the auxiliary source domain and target nighttime domain into a common latent space are highly necessary for effective nighttime optical flow adaptation.*

In this work, we explore a common-latent space as the intermediate bridge to mitigate the distribution misalignment between source domain and target nighttime domain in Fig. 1 (c). We introduce daytime image data and event camera data as two kinds of auxiliary domains to learn the intrinsic common spaces that contain the essential multi-source information associated with nighttime data. On one hand, the daytime image data contains dense visual features, which is definitely beneficial to effectively capture the global motion appearance. On the other hand, the event camera is an emerging neuromorphic vision sensor with high dynamic range (Gallego et al., 2019), which is specifically sensitive to the local motion boundary. Appearance knowledge and boundary knowledge are complementary to each other and important for nighttime optical flow.

Specifically, we propose a novel appearance-boundary domain adaptation framework for nighttime optical flow (ABDA-Flow) in Fig. 2. In appearance adaptation, we observe that the auxiliary daytime data and the nighttime data can be projected into a reflectance-aligned common space via the intrinsic image decomposition retinex model (Fu et al., 2016). The motion distributions of the two reflectance maps are very similar in the reflectance-aligned common space. We further map the aligned common features to their motion spaces, and encourage motion manifolds of both domains to be close to each other, thus benefitting consistently transferring global motion knowledge from daytime to nighttime domain. In boundary adaptation, we theoretically derive the motion correlation formula between paired nighttime frame and accumulated events within a spatiotemporal gradient-aligned common space. We figure out that the correlation of the two spatiotemporal gradient maps shares significant discrepancy measured by the Euclidean distance, benefitting us to contrastively transfer boundary knowledge from event to nighttime domain. Thus, appearance adaptation and boundary adaptation are perfectly complementary to each other, where the former is to transfer global motion appearance and the latter is to transfer local motion boundary. Overall, our main contributions are summarized:

- We propose a novel common space appearance-boundary adaptation framework for nighttime optical flow. To the best of our knowledge, this is the first work that leverages the common space adaptation learning for tackling the problem of feature representation misalignment in optical flow.

- We construct two common spaces: reflectance-aligned (appearance) common space between daytime and nighttime domains, and spatiotemporal gradient-aligned (boundary) common space between nighttime frame and accumulated events. Both appearance and boundary adaption are complementary to each other with better discriminative feature representations.

- We conduct extensive experiments on various datasets. Quantitative and qualitative results demonstrate that ABDA-Flow achieves state-of-the-art performance for nighttime optical flow.

## 2 RELATED WORK

**Optical Flow Estimation.** In recent years, CNN-based optical flow methods (Ranjan & Black, 2017; Ren et al., 2017; Sun et al., 2018; Stone et al., 2021; Teed & Deng, 2020) constructed cost volume and iterative strategies for optimizing optical flow, while transformer-based methods (Jiang et al., 2021; Huang et al., 2022; Lu et al., 2023) tokened 4D cost volume and incorporated transformer into optical flow estimation. However, these approaches usually suffer from extreme less-texture caused by the low dynamic range of conventional cameras, which weakens discriminative visual features, thus matching invalid motion features. On the contrary, event camera (Gallego et al., 2019) is an emerging neuromorphic vision sensor with high dynamic range, which can sense sparse motion boundary under nighttime scenes. Event-based optical flow methods (Gallego et al., 2018; Zhu et al., 2018; Paredes-Vallés & de Croon, 2021; Gehrig et al., 2021b) mainly follow frame-based framework, and train their networks via photometric constancy assumption. In this work, we leverage event camera to assist conventional camera to improve nighttime optical flow.

**Nighttime Optical Flow.** An intuitive solution is to perform visual enhancement with subsequent optical flow estimation. However, existing enhancement methods (Fu et al., 2016; Guo et al., 2016) are not designed for optical flow, and the possible enhanced results may lose visual features, thus contributing negatively to motion feature matching. Instead, a few methods have attempted to take an auxiliary domain to directly transfer knowledge to nighttime domain via domain adaptation in either input visual space or output motion space, *e.g.*, visual adaptation (Zheng et al., 2020) and motion adaptation (Li et al., 2021). Zheng et al. (2020) transformed daytime visual features to nighttime domain via noise model, and then estimated nighttime optical flow. Li et al. (2021) used gyroscope as the auxiliary domain to model the background motion features for assistantly improving nighttime motion features, while failed for independent objects. However, this direct adaptation falls short due to the large domain gap between auxiliary and nighttime domains, *e.g.*, features distribution misalignment. In this work, we employ daytime domain and event domain as the auxiliary domains, and explore two common-latent spaces as the intermediate bridges to directionally transfer global motion and local boundary knowledge to nighttime domain.

**Domain Adaptation.** Domain adaptation aims to tackle the distribution discrepancy between source and target domains. Degraded scene understanding can be formulated as a domain adaptation problem, which focuses on transferring specific knowledge from source clean domain to target degraded domain. Existing domain adaptation methods under degraded scenes mainly directly transfer knowledge in visual space (Chen et al., 2021; Gao et al., 2022) or task space (Liu et al., 2021). However, we discover that there exists a large domain gap due to the intrinsic heterogeneous nature of the feature representations between clean and degraded domains, limiting these direct adaptation methods. To overcome this issue, we explore a common-latent space as the intermediate bridge to reinforce feature alignment. Note that, common space adaptation can be applied for any degraded scene understanding tasks, and the key is to find the appropriate common space according to the specific task.

## 3 COMMON APPEARANCE-BOUNDARY ADAPTATION

### 3.1 OVERALL FRAMEWORK

Nighttime optical flow is formulated as a task of exploring the common-latent space to transfer knowledge from source auxiliary (e.g., daytime image and event) to target domain (e.g., nighttime image). To reinforce the feature alignment between source and target domains, we build two common-latent spaces. In this work, we propose a novel common appearance-boundary adaptation framework to learn the two intermediate common-latent spaces. As shown in Fig. 2, the whole model structure looks complicated but is simply modularized into four sub-modules, where two of them are used to build the common space (e.g., common latent reflectance space and common latent boundary space) and the other two are used for motion adaptation (motion distribution alignment and motion boundary contrastive). The common latent reflectance space module and motion distribution alignment module make up the appearance adaptation to consistently transfer motion appearance knowledge from

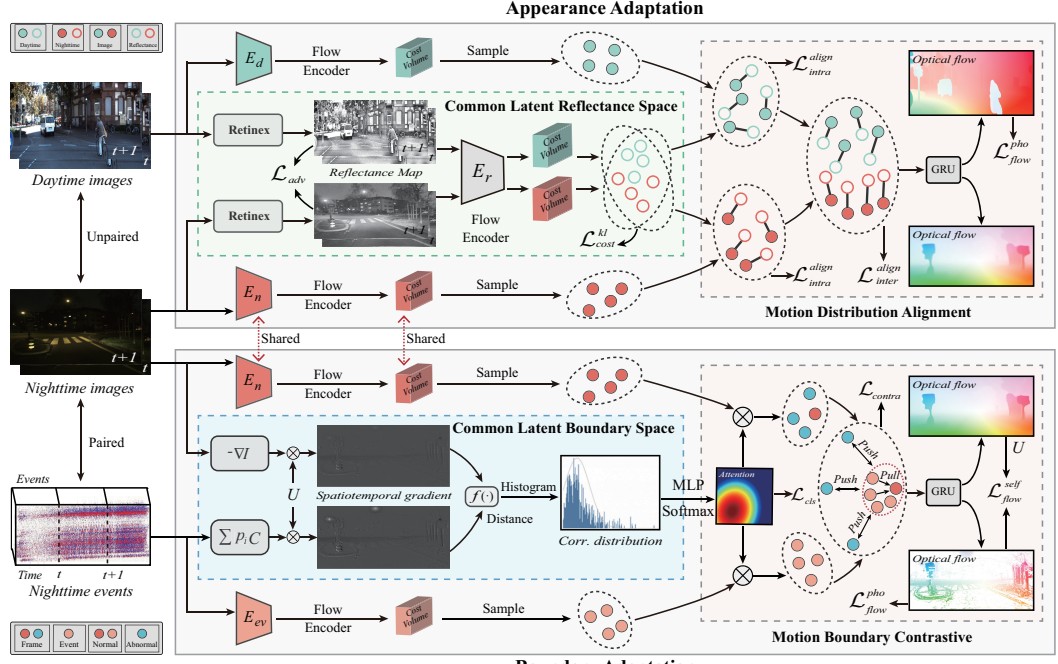

Figure 2: The architecture of the ABDA-Flow mainly contains appearance and boundary adaptation. In appearance adaptation, we take retinex model to align daytime and nighttime images into the reflectance-aligned common space. We then map the common features to motion space, and make the motion distributions between daytime and nighttime domains aligned. In boundary adaptation, we transform nighttime image and event stream to the spatiotemporal gradient-aligned common space. We then calculate the correlation statistic between the two spatiotemporal gradient maps to generate an attention map for guiding the boundary features alignment between nighttime and event domains.

daytime to nighttime domain, while the common latent boundary space module and motion boundary contrastive module make up the boundary adaptation to transfer local boundary knowledge from event to nighttime domain. Under this unified framework, the common appearance and boundary adaptation complement each other, and jointly transfer the dominant knowledge to nighttime domain.

## 3.2 COMMON APPEARANCE ADAPTATION

Estimating optical flow from nighttime images is difficult since nighttime degradations break the brightness constancy assumption, which most of optical flow methods rely on. We argue that scene motion is not affected by illumination, but by the intrinsic appearance of the scene. Therefore, we aim to explore a common-latent space robust to illumination to associate daytime and nighttime domains.

**Common Reflectance Space.** Retinex model (Fu et al., 2017; Wu et al., 2022) assumes that a image can be intrinsically decomposed into illumination $L$ and reflectance $R$, where reflectance represents image appearance. Motivated by this, we argue that daytime and nighttime reflectance of the same scene should be consistent. This makes us naturally consider how different the optical flows obtained by daytime and nighttime reflectance maps are. To illustrate this, we map daytime and nighttime images and reflectance maps to the same motion manifold. In Fig. 3, we have a key observation: *motion distributions from daytime and nighttime reflectance maps are similar*. This motivates us to take reflectance as the common-latent space to associate daytime and nighttime domains.

According to the retinex model $I = R \cdot L$, given the daytime frames $[I_d^t, I_d^{t+1}]$ and the nighttime frames $[I_n^t, I_n^{t+1}]$, we decompose them to obtain the corresponding reflectance maps, namely $[R_d^t, R_d^{t+1}, R_n^t, R_n^{t+1}]$, where $d, n$ denote the daytime and nighttime. Note that, the retinex architecture of our framework is similar to Uretinex-net (Wu et al., 2022), which is pre-trained on the public datasets (Wu et al., 2022) for better initialization. To further make the daytime and nighttime reflectance maps look similar, we enforce the adversarial loss (Zhu et al., 2017):

$$\mathcal{L}_{adv} = \mathbb{E}_d[logD(R_d)] + \mathbb{E}_n[log(1 - D(R_n))], \tag{1}$$

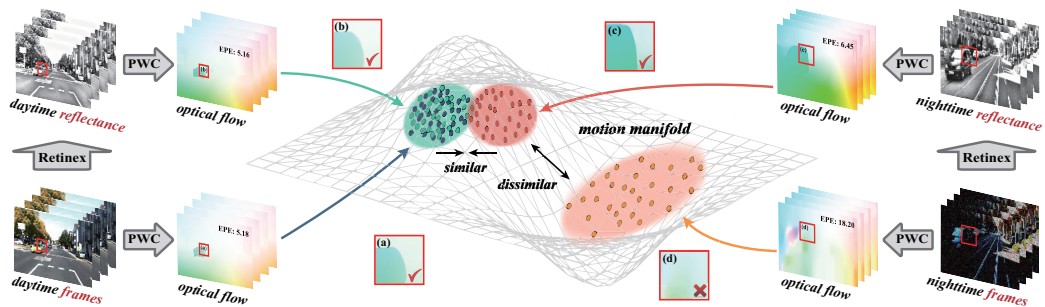

Figure 3: Motion distribution of daytime and nighttime domains. Optical flow of nighttime frame suffers degradation while flow of daytime frame is sharp. Motion distribution of nighttime reflectance is similar to those of daytime frame and reflectance, but dissimilar to that of nighttime frame. This motivates us to take reflectance as the common latent space to transfer knowledge.

where $D$ is the discriminator. Then, we employ the motion feature extractor $E_r$ of reflectance to encode the reflectance maps to the cost volume space $[cv_d^r, cv_n^r]$, where $r$ denotes the reflectance. Note that, cost volume stores the correlation value between adjacent frames, which is formulated as $cv = (f_t)^T \cdot w(f_{t+1})$, where $f$ is the visual features, $T$ denotes transpose operator and $w$ is the warp operator. We further align the cost volumes of daytime and nighttime reflectance maps to guarantee the consistent motion distribution using K-L divergence with softmax function $\Phi$:

$$\mathcal{L}_{cost}^{kl} = \sum \Phi(cv_n^r) \cdot log \frac{\Phi(cv_n^r)}{\Phi(cv_d^r)}. \tag{2}$$

**Motion Distribution Alignment.** To ensure the daytime→nighttime directional knowledge transfer, we take the cost volume of the common reflectance-aligned space to associate motion features of daytime and nighttime domains. We divide the transfer process into two phases: intra-domain motion alignment which transfers knowledge between visual-based and reflectance-based motion spaces within the same domain, and inter-domain motion alignment which is the cross-domain knowledge transfer. We first train the daytime optical flow network with photometric loss (Yu et al., 2016):

$$\mathcal{L}_{flow}^{pho} = \sum \psi(I_d^t - w(I_d^{t+1})) \odot (1-O) / \sum (1-O), \tag{3}$$

where $\psi$ is a sparse $L_p$ norm ($p = 0.4$). $O$ is the occlusion mask by checking forward-backward consistency (Zou et al., 2018), and $\odot$ is a matrix element-wise multiplication. As for the intra-domain motion alignment, we enforce the pixel-level cost consistency loss:

$$\mathcal{L}_{intra}^{align} = ||cv_d - cv_d^r||_1 + ||cv_n - cv_n^r||_1. \tag{4}$$

After that, we calculate the cost volume discrepancy between visual-based and reflectance-based motion spaces within the same domain, and use K-L divergence to constrain the distribution discrepancy between daytime and nighttime domains to achieve the inter-domain motion alignment:

$$\mathcal{L}_{inter}^{align} = \sum \Phi(cv_n - cv_n^r) \cdot log \frac{\Phi(cv_n - cv_n^r)}{\Phi(cv_d - cv_d^r)}. \tag{5}$$

Hence, the common appearance adaptation first transfers the global motion knowledge from daytime to nighttime in common reflectance space via $\mathcal{L}_{cost}^{kl}$, then enforce the pixel-level motion consistency between reflectance to image in daytime and nighttime individually via $\mathcal{L}_{intra}^{align}$, and finally, transfer the motion residual between image and reflectance from daytime to nighttime via $\mathcal{L}_{inter}^{align}$, in which it further diminishes the small motion errors in the nighttime.

### 3.3 COMMON BOUNDARY ADAPTATION

Appearance adaptation can promise a preliminary result for nighttime optical flow, while limited by low dynamic range of conventional cameras under nighttime scenes, there exist weakened visual features, thus matching the inaccurate motion features. This problem cannot be solved by appearance adaptation alone. According to our investigation, event camera (Gallego et al., 2019) has the advantage of high dynamic range which promises sensing motion boundary. Therefore, we introduce the event camera (event domain) to assist the conventional camera in improving the motion boundary. However, there are two difficulties: spatial alignment and large domain gap between nighttime image and event.

**Paired Image and Event.** The spatially aligned nighttime image and event can be obtained in two ways (seeing supplementary for details). First, we have set up a physically coaxial optical system

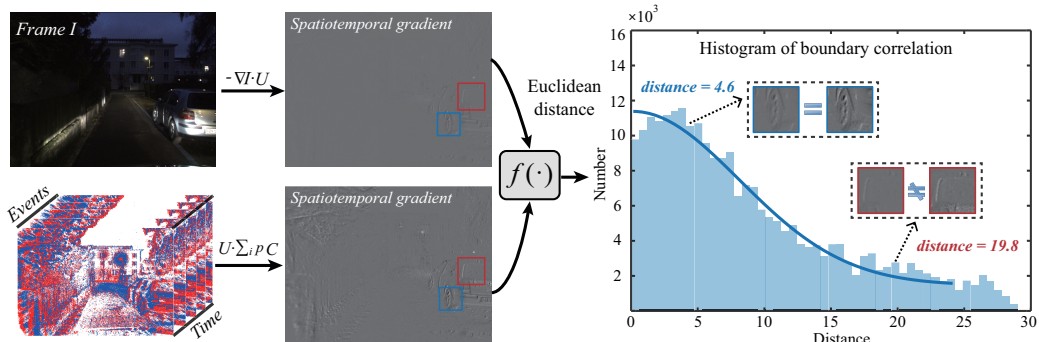

Figure 4: Motion correlation statistic between nighttime image and event domains. We use Euclidean distance to calculate the motion correlation between the nighttime image and accumulated events within the spatiotemporal gradient common space. The larger the distance is, the larger correlation discrepancy is, and the more dissimilar the boundaries of the two spatiotemporal gradient maps. This motivates us to contrastively transfer boundary knowledge to nighttime image domain.

with a beam splitter for the event and image sensor, in which the two modalities are inherently spatial aligned by the shared light path. Second, we start from the spatial alignment algorithm perspective by performing a standard stereo rectification, and then fine-tune the registration error by pixel offset (Tulyakov et al., 2022), which can ensure a reliable paired nighttime image and event.

**Common Boundary Space.** To explore the common space for the image-event domain gap, we ideally extend the optical flow basic model (Paredes-Vallés & de Croon, 2021) via Taylor expansion:

$$I(x, t) = I(x, t) + (\frac{dx}{dt}\frac{\partial I}{\partial x} + \frac{\partial I}{\partial t}) + O(dx, dt). \tag{6}$$

where $u$ is estimated motion. We remove the high-order error term $O(dx, dt)$ to approximate Eq. 6:

$$I_t = -\nabla I \cdot U, \tag{7}$$

where $U$ denotes optical flow estimated by adjacent frames and $\nabla I$ is spatial gradient field of the frame. Note that, optical flow $U$ is firstly pre-trained via appearance adaptation on daytime images for accuracy since the flow encoders of appearance and boundary adaptation for nighttime domain are shared. $I_t$ denotes brightness change along the time dimension, which can be approximated as the accumulated events warped by the optical flow in a certain time window, namely $I_t = \Delta L_{ev} = U \cdot \sum_{e_i \in \Delta t_k} p_i C$. And we further transform Eq. 7 as follows:

$$\Delta L_{ev} = U \cdot \sum_{e_i \in \Delta t_k} p_i C = -\nabla I \cdot U, \tag{8}$$

where $-\nabla I \cdot U$ and $\Delta L_{ev}$ are both the spatiotemporal gradient. $e_i$ is the event timestamp, $p_i$ is the event polarity, and $C$ denotes the event trigger threshold. Eq. 8 indicates that the spatiotemporal gradient of image and event domains are consistent, and can serve as the spatiotemporal gradient-aligned common space, namely common latent boundary space. When applied for nighttime scenes, conventional camera suffers from weakened texture, while the event camera still has clear boundaries. We use Euclidean distance to construct a pixel-wise motion correlation metric formula to measure the discrepancy between nighttime image and event domains within the common boundary space:

$$Corr = N(f(\Delta L_{ev}, -\nabla I \cdot U)), \tag{9}$$

where $f$ is correlation metric with Euclidean distance, and $N$ is normalization. In Fig. 4, we further obtain the correlation statistic distribution via histogram (corresponding to the histogram in Fig. 2), and figure out that *the correlation of the spatiotemporal gradient maps calculated from nighttime images and accumulated events shares significant discrepancy*. This motivates us to utilize the common boundary space as an intermediate bridge to contrastively transfer boundary knowledge.

**Motion Boundary Contrastive.** Within the spatiotemporal gradient-aligned common space, the motion correlation can perceive the degree of nighttime degradation in different regions. Thus, through MLP and softmax, we take the correlation statistic distribution as a prior to generate a weight attention map $A$ to classify the motion classes with the cross-entropy loss:

$$\mathcal{L}_{cls} = -\mathbb{E} \sum_{i,j \in A} \sum_{k=0}^{K} \mathbb{I}_{[k=\mathbf{y}]} log(A(i, j)), \tag{10}$$

where motion classes contain $K$ manually defined motion features that can reflect different degrees of degradation, which is determined by the correlation statistical distribution. 0 corresponds to normal

Table 1: Quantitative results on synthetic (Noise) Dark-KITTI2015 (D-KITTI / ND-KITTI) datasets.

| Method | | DarkFlow-PWC | Selflow | | | SMURF | | | ABDA |
|--------|--------|--------------|---------|-----------|-----------|--------|-----------|-----------|------|
| | | | − | (KinD++) + | AGLLNet + | − | (KinD++) + | AGLLNet + | |
| D-KITTI | EPE | 7.56 | 14.22 | 12.58 | 11.70 | 11.36 | 10.03 | 8.42 | **3.47** |
| | Fl-all | 35.75% | 55.87% | 48.69% | 46.31% | 45.88% | 44.65% | 39.25% | **16.13%** |
| ND-KITTI | EPE | 8.56 | 18.01 | 16.75 | 14.54 | 13.40 | 11.95 | 10.26 | **4.35** |
| | Fl-all | 41.28% | 65.43% | 59.55% | 55.26% | 54.21% | 45.91% | 45.60% | **23.86%** |

(a) Nighttime Images     (b) SMURF     (c) DarkFlow-PWC     (d) GMA     (e) ABDA-Flow

Figure 5: Visual comparison of optical flows on real nighttime images of DSEC dataset.

motion. $\mathbf{y}$ is the pre-classified motion class label, and $\mathbb{I}_{[k=\mathbf{y}]}$ is an indicator function. Next, we multiply the cost volume of nighttime image domain and that of event domain with the attention map to acquire the motion features with category attributes. To get rid of the degraded motion features, we exploit contrastive learning to pull normal motion features together while push abnormal motion features away. We argue that motion features sampled from event domain should be accurate as positives $f_{P_k^{ev}}$; motion features whose class index is $0$ are the positive samples $f_{P_j^n}$, and those corresponding to other classes are the negatives $f_{N_i^n}$, sampled from nighttime image domain. We align the normal motion features of nighttime image domain and event domain via contrastive adaptation loss:

$$\mathcal{L}_{contra} = \frac{1}{N} \sum_{k=1}^{N} \sum_{j=1}^{N} \frac{exp(f_{P_j^n} \cdot f_{P_k^{ev}}/\tau)}{exp(f_{P_j^n} \cdot f_{P_k^{ev}}/\tau) + \sum_{i=1}^{N} exp(f_{N_i^n} \cdot f_{P_j^n}/\tau)}, \quad (11)$$

where $N$ denotes the positive/negative sample number, $\tau$ is the scale parameter. We then estimate the nighttime motion $F_n$ and event motion $F_{ev}$ with the aligned features via the motion consistency loss:

$$\mathcal{L}_{flow}^{self} = \sum ||F_n - F_{ev}||_1 \odot V / \sum V, \quad (12)$$

where $V$ represents the valid motion mask, obtained from the attention map $A$ via threshold segmentation, where the threshold is the probability value corresponding to the motion class $0$. Note that, we also train the event optical flow model with the photometric loss $\mathcal{L}_{flow}^{pho}$.

### 3.4 OPTIMIZATION

Consequently, the total objective for the proposed framework is written as follows:

$$\mathcal{L}_{ABDA} = \mathcal{L}_{flow}^{pho} + \lambda_1 \mathcal{L}_{adv} + \lambda_2 \mathcal{L}_{cost}^{kl} + \lambda_3 \mathcal{L}_{intra}^{align} + \lambda_4 \mathcal{L}_{inter}^{align} + \lambda_5 \mathcal{L}_{cls} + \lambda_6 \mathcal{L}_{contra} + \lambda_7 \mathcal{L}_{flow}^{self}. \quad (13)$$

The first term is to initialize the daytime optical flow and event optical flow networks, the second and third terms are to constrain the common reflectance alignment, the fourth and fifth terms are to transfer global motion knowledge from daytime to nighttime domain, and the intention of the sixth term is to generate the attention map using the correlation statistic distribution within the common boundary space, the last two terms aim to transfer the local boundary knowledge from event to nighttime image domain. $[\lambda_1, ..., \lambda_7]$ are the weights that control the importance of the related losses.

## 4 EXPERIMENTS

### 4.1 EXPERIMENT SETUP

**Dataset.** We conduct extensive experiments on synthetic and real datasets. The synthetic dataset is synthesized by the noise model (Zheng et al., 2020) on KITTI2015 (Menze & Geiger, 2015), named as (noise) Dark-KITTI2015. The real datasets include the public datasets (*e.g.,* Dark-GOF and Dark-DSEC) and the proposed low light frame-event (LLFE) dataset. Dark-GOF and Dark-DSEC are the nighttime parts of GOF (Li et al., 2021) and DSEC (Gehrig et al., 2021a). LLFE covers various nighttime illumination conditions. Note that, Dark-DSEC and LLFE are the paired frame-event datasets obtained via stereo rectification and coaxial optical system, respectively.

Table 2: Quantitative results on real nighttime datasets. '–' denotes none of the training data.

| Method | | Frame-based methods | | | | | | Event-based methods | | |
|---|---|---|---|---|---|---|---|---|---|---|
| | | SMURF | PWC | GMA | DarkFlow-PWC | GyroFlow | **ABDA** | EV-FlowNet | E-RAFT | **E-ABDA** |
| Dark-GOF | EPE | 2.87 | 10.26 | 1.14 | 3.35 | 0.92 | **0.85** | – | – | – |
| | Fl-all | 28.20% | 65.47% | 13.60% | 32.18% | 9.85% | **7.94%** | – | – | – |
| Dark-DSEC | EPE | 2.13 | 3.08 | 1.48 | 2.29 | – | **0.74** | 3.21 | 0.82 | **0.78** |
| | Fl-all | 36.75% | 59.32% | 24.10% | 38.23% | – | **11.85%** | 63.25% | 13.81% | **12.26%** |

Figure 6: Visual comparison of optical flows on the proposed unseen LLFE with various illumination.

**Implementation Details.** We set the sample number $N$ as 1000 and the motion class number $K$ as 10. During the training phase, we need only three steps. First, we train daytime and event optical flow models to ensure that the proposed framework can learn accurate motion knowledge. Second, we train nighttime optical flow model via appearance adaptation to transfer motion knowledge from daytime to nighttime domain. Finally, we use boundary adaptation to further train nighttime optical flow model for transferring motion knowledge from event to nighttime image domain. During the testing phase, we only need the single nighttime optical flow model for inference. We choose the average end-point error (EPE (Dosovitskiy et al., 2015)) and the lowest percentage of erroneous pixels (Fl-all (Menze & Geiger, 2015)) as the evaluation metrics for the quantitative evaluation.

**Comparison Methods.** We choose visual adaptation DarkFlow-PWC (Zheng et al., 2020) and motion adaptation GyroFlow (Li et al., 2021) for a fair comparison. Several supervised (PWC-Net (Sun et al., 2018) and GMA (Jiang et al., 2021)) and unsupervised (Selflow (Liu et al., 2019) and SMURF Stone et al. (2021)) methods are also compared. Note that, the supervised methods are first trained on Dark-KITTI2015, and then trained on target real nighttime datasets via self-supervised learning (Stone et al., 2021). For the comparison on synthetic datasets, we have two training strategies for competing methods, one is to directly train on nighttime images; and the other is the two-stage one by performing image enhancement first (*e.g.*, KinD++ (Zhang et al., 2019), AGLLNet (Lv et al., 2021)), and then train them on the enhanced results (named as (KinD++)+ / AGLLNet+). In addition, we also compare with the event optical flow methods (*e.g.*, EV-FlowNet (Zhu et al., 2018) and E-RAFT (Gehrig et al., 2021b)) with our event optical flow model on the DSEC and LLFE datasets.

## 4.2 COMPARISON EXPERIMENTS

**Comparison on Synthetic Datasets.** In Table 1, we choose unsupervised methods for fair comparison. We can conclude that the proposed method significantly outperforms the competing methods by a large margin. Note that, the pre-processing image enhancement approaches do benefit to nighttime optical flow, while the performance is still limited due to the possible artifacts after enhancement. Compared with DarkFlow-PWC, the proposed method with the latent-space adaptation works better.

**Comparison on Real Datasets.** In Table 2, the unsupervised method suffers degradations, and the supervised methods cannot work well due to the gap between synthetic and real images. There exist outliers in the results of DarkFlow-PWC and GyroFlow that are designed for nighttime scenes in Fig. 5. In contrast, our result is sharp-boundary. Moreover, we compare with SOTA event optical flow methods in Table 2, demonstrating the superiority of the proposed event optical flow model.

**Generalization for Unseen Nighttime Scenes.** In Fig. 6, we choose the proposed LLFE dataset as the unseen nighttime scenes for generalization comparison. Under real nighttime conditions with various illumination, the proposed method performs better than other competing methods. The main reason is that the proposed common space adaptation can explicitly learn the intrinsic motion knowledge for nighttime scenes, regardless of the illumination change.

Table 3: Discussion on effect of common space.

| Strategy | EPE |
|---|---|
| w/o motion/reflect./bound. | 1.51 |
| w/ motion, w/o reflect./bound. | 1.09 |
| w/ motion/reflect., w/o bound. | 0.87 |
| w/ motion/bound., w/o reflect. | 0.95 |
| w/ motion/reflect./bound. | **0.74** |

Table 4: Ablation study on adaptation losses.

| $\mathcal{L}_{intra}^{align}$ | $\mathcal{L}_{inter}^{align}$ | $\mathcal{L}_{contra}$ | $\mathcal{L}_{flow}^{self}$ | EPE |
|---|---|---|---|---|
| × | × | × | × | 1.45 |
| ✓ | × | × | × | 1.24 |
| ✓ | ✓ | × | × | 1.05 |
| ✓ | ✓ | ✓ | × | 0.85 |
| ✓ | ✓ | ✓ | ✓ | **0.74** |

Table 5: Discussion on training data and optical flow backbone.

| Training data | Method | EPE |
|---|---|---|
| daytime, nighttime | CycleGAN + our baseline | 1.41 |
| | Our appearance adaptation | 0.87 |
| daytime, nighttime, event | CycleGAN + our baseline + E-RAFT | 1.33 |
| | Ours w/ CNN backbone | 0.77 |
| | Ours w/ Transformer backbone | **0.74** |

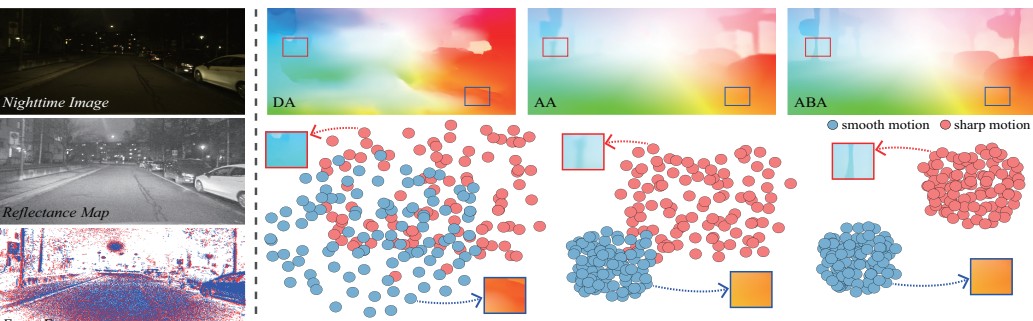

Figure 7: t-SNE of motion distribution in smooth and sharp motion regions during appearance and boundary adaptation. Direct adaptation cannot work well. Appearance adaptation with reflectance could smooth global motion. Boundary adaptation with event further sharpens the motion boundary.

## 4.3 ABLATION STUDY AND DISCUSSION

**How does Common Space Work?** In Table 3, we demonstrate the role of common space on knowledge transfer. Motion adaption is a motion transfer part of appearance adaptation and boundary adaptation, which does improve nighttime optical flow, while the performance has an upper limit due to the domain gap. Common space further contributes positively to the optical flow result. The main reason is that common space can bridge the domain gap, and guide directional knowledge transfer.

**Direct Adaptation *v.s.* Common Space Adaptation.** In Fig. 7, we visual the motion results and low-dimension feature distributions in smooth and sharp motion regions via t-SNE. Direct adaptation cannot work well, and the entire distribution is cluttered. Common appearance adaptation with reflectance concentrates smooth motion features, and common boundary adaptation with event further clusters sharp motion features. Clustered low-dimension feature distributions illustrate that the common appearance-boundary adaptation could effectively learn discriminative motion features.

**Effectiveness of Adaptation Losses.** In Table 4, $\mathcal{L}_{intra}^{align}$ and $\mathcal{L}_{inter}^{align}$ significantly improve nighttime optical flow performance, since appearance adaptation could transfer the dense motion knowledge to nighttime domain. $\mathcal{L}_{contra}$ and $\mathcal{L}_{flow}^{self}$ slightly further improve the optical flow result, because boundary adaptation mainly transfers sparse boundary knowledge to nighttime image domain.

**Influence of Training Data and Backbone.** In Table 5, we replace common space adaptation with CycleGAN (Zhu et al., 2017) and CNN-based RAFT for comparison on multi-source training data and optical flow backbone. For training data, the "CycleGAN+" strategies do not perform as well as common space adaptation. For the backbone, the performances of transformer-based and CNN-based backbones are almost the same. Therefore, training data diversity and backbone can indeed improve the performance, while common space adaptation is the key to solving nighttime optical flow.

## 5 CONCLUSION

In this work, we propose a novel common appearance-boundary adaptation framework to learn an intermediate common space with discriminative feature representations for nighttime optical flow. To the best of our knowledge, we are the first to investigate the new common space learning paradigm for tackling the heterogeneous feature representations due to the gap between auxiliary and nighttime domains for nighttime optical flow. We construct the daytime-nighttime reflectance-aligned common space and the image-event spatiotemporal gradient-aligned common space. The two common space adaptations are complementary to each other for joint global appearance and local boundary motion estimation. We demonstrate that the proposed method significantly outperforms the state-of-the-art methods. We believe that our work could not only facilitate the development of the nighttime optical flow but also enlighten the researchers of the broader field, i.e., adverse scene understanding.

## ACKNOWLEDGMENTS

This work was supported in part by the National Natural Science Foundation of China under Grant 62371203. The computation is completed in the HPC Platform of Huazhong University of Science and Technology.

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
