# OpenReview forum: "Exploring the Common Appearance-Boundary Adaptation for Nighttime Optical Flow"
_ICLR.cc/2024/Conference — ICLR 2024 spotlight_

### Official Review · Reviewer_R9ad · 2023-10-26

**Soundness:** 3 good
**Presentation:** 4 excellent
**Contribution:** 4 excellent
**Rating:** 10
**Confidence:** 3

**Summary:**

The study addresses nighttime optical flow, hampered by low texture and high noise. Traditional methods employ domain adaptation from auxiliary to nighttime domains but struggle due to domain gaps. To overcome this, the study introduces a common-latent space, using daytime and event domains. It shows motion appearance knowledge can be transferred effectively in reflectance-aligned spaces, and theoretical derivations emphasize substantial correlations between spatiotemporal gradients. Appearance and boundary adaptation are complementary and effectively transfer global motion and local boundary knowledge to nighttime domains, as validated by extensive experiments.

**Strengths:**

1) The paper is clearly written and easy to follow.
2) The novelty of this paper, in my opinion, is significant to the community. The concept of “common space adaptation” is very effective to reinforce feature alignment between domains, and it has great potentials to be applied for any degraded scene understanding tasks.
3) The constructed common spaces are from both appearance (reflectance) and boundary (gradient) sides. They are complementary and sound reasonable for the performance improvement.

**Weaknesses:**

While the proposed method appears promising, the complexity of the loss function involving seven balance weights raises concerns. Identifying the appropriate values for these seven weights efficiently is a challenge. Performing a grid search to determine these values might be time-consuming and resource-intensive.

**Questions:**

Please refer to the weakness part.

---

> ### Author Response · Authors · 2023-11-15
>
> **Q1:** Complexity of determining the balance weights for the loss function.
>
> **A:** We deeply thanks for your strong recommendation of the proposed framework. We do acknowledge that, if the whole framework is directly trained together, determining the balance weights of the loss function can be time-consuming. However, in Fig. 2 of this paper, each component has its own specific physical meaning. In appearance adaptation, the common latent reflectance space bridges the daytime-nighttime domain gap, and assists the motion distribution alignment module in transferring motion appearance knowledge from daytime to nighttime domain. In boundary adaptation, the common latent boundary space closes the event-image domain gap, thus guiding the motion boundary contrastive module to transfer local boundary knowledge from event to nighttime image domain. Therefore, the whole framework can be divided into different components for separate training (seeing the "Implementation Details" part of this paper). When each module is trained separately, only 1-2 balance weights need to be determined, which is easy to adjust. In the final joint optimization, we fine-tune the whole framework using the preset balance weights. In addition, the weight sensitivity of the loss function has been analyzed in the "Implementation of Training Details" subsection of the proposed supplementary material, where we have discussed four balance weights of the losses that are related to domain adaptation. We will further add the weight sensitivity experiments of the other balance weights for the loss function in the revised manuscript.

---

> ### Author Response · Authors · 2023-11-19
>
> Dear Reviewer R9ad,
>
> Thank you for your recognition and recommendation of our work. We have explained the question raised by you. We are expecting for your reply.
>
> Best wishes.

---

> ### Author Response · Authors · 2023-11-21
>
> Dear Reviewer R9ad,
>
> We express our sincere gratitude for your recognition and recommendation of our work. We have actively discussed your valuable queries. And, we eagerly anticipate your prompt feedback.
>
> Best wishes.

---

> > ### Comment · Reviewer_R9ad · 2023-11-22
> >
> > Dear Authors：
> >
> > I've read the feedback and that addresses my concerns. I would maintain the initial scores.
> >
> > Good Luck!

---

> > > ### Author Response · Authors · 2023-11-22
> > >
> > > Dear Reviewer R9ad,
> > >
> > > It's an honor to receive your feedback and thank you again for your recognition of our work.
> > >
> > > Best wishes.

---

### Official Review · Reviewer_hzUU · 2023-10-31

**Soundness:** 3 good
**Presentation:** 4 excellent
**Contribution:** 3 good
**Rating:** 8
**Confidence:** 2

**Summary:**

This paper  focuses on the nighttime optical flow task, which is a challenging.  This paper  expolit two auxiliary daytime adn event domains, and present a common apperance-boundary adaption framework for nighttime optical flow. For apperance and boundary adaption, this paper have the new exploration, and both them are complementary to each other.  Extensive experiments  are conducted on various datasets, showing the SOTA performance.

**Strengths:**

1. This paper is well-presented, including the figures and tables.

2. The experiments are well-conducted, including  main comparsions, ablation studies and viual results. The method achieves the sota performance.

3. From the provided video demo, the proposed method genelizes well across many scenes.

**Weaknesses:**

It would be better to add some analyses on the failure cases and show the limitations of the method and give some discussions.

**Questions:**

Hope authors to release the codes to benefit the community on this field.

---

> ### Author Response · Authors · 2023-11-15
>
> **Q1:** Limitation of the proposed method.
>
> **A:** Thanks for appreciating our work. We have specifically discussed the limitation of the proposed method in the "3.7 Limitation'' subsection of the supplementary material. The proposed method can model the x, y-axis motion pattern, but fails to estimate the radial motion along the z-axis. The key to z-axis radial motion estimation is to obtain the accurate scene depth information. From the perspective of imaging mechanism, the frame camera records absolute luminance in the x, y-axis via global scan, while the event camera mainly triggers x, y-axis events due to brightness change, and is difficult to respond to z-axis motion in a short time. Therefore, it is difficult for frame and event cameras to directly perceive z-axis motion information. For example, when imaging that a vehicle is approaching us from far to near along the center of the camera, the frame camera cannot capture the relative motion of the vehicle, and the event camera also fails to trigger the corresponding events. From the perspective of knowledge transfer, the common space we constructed follows the x, y-axis basic optical flow model, serving as an intermediate bridge to guide the knowledge transfer, which lacks the constraint on z-axis motion knowledge. Both aspects limit the ability of the proposed method to model z-axis radial motion. In the future, we will further introduce LiDAR which is robust to illumination, to directly measure the depth information to assistantly estimate the 3D motion.

---

> ### Author Response · Authors · 2023-11-19
>
> Dear Reviewer hzUU,
>
> Thank you very much for reviewing our work. We have discussed the limitation raised by you. We are looking forward to your response.
>
> Best wishes.

---

> ### Author Response · Authors · 2023-11-21
>
> Dear Reviewer hzUU,
>
> Thank you for your detailed review. We have taken your valuable feedback into consideration and made some discussions to address your concerns about the limitation. As the discussion deadline is approaching, we eagerly await your timely response.
>
> Best wishes.

---

### Official Review · Reviewer_koka · 2023-11-01

**Soundness:** 3 good
**Presentation:** 3 good
**Contribution:** 3 good
**Rating:** 6
**Confidence:** 4

**Summary:**

This paper proposes a novel common appearance-boundary adaptation framework for nighttime optical flow estimation. The authors explore a common latent space as an intermediate bridge to reinforce feature alignment between auxiliary and nighttime domains. They construct two common spaces: a reflectance-aligned common space between daytime and nighttime domains, and a spatiotemporal gradient-aligned common space between nighttime frame and accumulated events. The appearance adaptation transfers global motion knowledge from daytime to nighttime domain, while the boundary adaptation transfers local motion boundary knowledge from event to nighttime domain. The proposed method, ABDA-Flow, achieves state-of-the-art performance for nighttime optical flow.

**Strengths:**

1. The common latent space approach effectively mitigates the distribution misalignment issue between source and target domains.
2. The appearance and boundary adaptations complement each other, jointly transferring global motion and local boundary knowledge to the nighttime domain.
3. The proposed method achieves state-of-the-art performance for nighttime optical flow estimation.

**Weaknesses:**

1. The method may be more complex to implement and train compared to simpler optical flow estimation techniques.
2. The effectiveness of the proposed method may be limited to specific nighttime optical flow tasks and datasets.
3. The runtime of the method may be slower than some other optical flow estimation techniques due to the Transformer architecture.

**Questions:**

1. How does the proposed common appearance-boundary adaptation framework compare to other domain adaptation techniques in terms of computational complexity and efficiency?
2. Can the proposed method be applied to other challenging optical flow estimation tasks, such as low-light or high-speed scenarios?
3. How does the proposed method handle the trade-off between model size and computational efficiency? Are there any plans to further optimize the runtime or explore other efficient optical flow estimation techniques?

---

> ### Author Response · Authors · 2023-11-15
>
> **Q1:** How does the proposed common appearance-boundary adaptation framework compare to other domain adaptation techniques in terms of computational complexity and efficiency?
>
> **A:** Thanks for your recognition of our work. Compared with other domain adaptation techniques, the proposed common appearance-boundary adaptation framework only introduces additional computational consumption in the training phase, but can greatly improve the optical flow performance. During the inference phase, the final model is the same as the typical optical flow backbone, which only needs the nighttime optical flow encoder $E_n$ and $GRU$ in Fig. 2 of this paper for testing. To fairly compare the computational complexity and efficiency, with the same optical flow backbone and training data, we compare the proposed common space adaptation with the visual adaptation ("CycleGAN+our flow baseline" in Table 5 of this paper) and the motion adaptation ("CycleGAN+our flow baseline+E-RAFT" in Table 5 of this paper) in complexity and performance as shown in the following table:
>
> |       Training data       |                Method                 | FLOPs (G) | Parameters (M) |   EPE    |
> | :-----------------------: | :-----------------------------------: | :-------: | :------------: | :------: |
> |    daytime, nighttime     |      CycleGAN+our flow baseline       |   1.85    |     31.28      |   1.41   |
> |                           |     Common appearance adaptation      |   1.87    |     35.84      | **0.87** |
> | daytime, nighttime, event |   CycleGAN+our flow baseline+E-RAFT   |   2.58    |     46.72      |   1.33   |
> |                           | Common appearance-boundary adaptation |   2.42    |     56.26      | **0.74** |
>
> ​	The results indicate that the complexity (i.e., FLOPs and Parameters) does not increase much, but the optical flow performance (i.e., EPE) is significantly improved. Therefore, the proposed common space adaptation only sacrifices a slight computational cost to transfer knowledge in the training stage, but can greatly improve the nighttime optical flow in the inference stage.
>
> **Q2:** The impact of common space adaptation on the model size and efficiency of the optical flow model for inference.
>
> **A:** It should be emphasized that the focus of this work is not the single optical flow backbone in the inference stage, but the knowledge transfer framework in the training stage for the nighttime optical flow task. The final optical flow model only needs the nighttime optical flow encoder $E_n$ and $GRU$ in Fig. 2 of this paper for testing, which can be replaced with any optical flow backbone. The model size and runtime are related to the optical flow backbone, while the accuracy of nighttime optical flow mainly depends on the knowledge transfer framework during the training phase. In the "3.5 Inference Time'' subsection of the proposed supplementary material, we have compared the runtime and performance of different optical flow backbones that before and after being trained under the proposed framework. In the table below, we further select the latest lightweight optical flow network EMD-S [1] to compare the model size, runtime, and accuracy without and with knowledge transfer:
>
> |              Method               | Model size (M) | Runtime (ms) | EPE  |
> | :-------------------------------: | :------------: | :----------: | :--: |
> |                GMA                |      5.9       |      79      | 1.48 |
> |     GMA w/ common adaptation      |      5.9       |      79      | 0.79 |
> |            Transformer            |      18.2      |     273      | 1.43 |
> | Transformer  w/ common adaptation |      18.2      |     273      | 0.74 |
> |               EMD-S               |      4.5       |      42      | 1.68 |
> |    EMD-S w/ common adaptation     |      4.5       |      42      | 0.81 |
>
> ​	We have two conclusions. First, model size and runtime are only related to the optical flow backbone. Second, the proposed common space adaptation framework can significantly improve the nighttime optical flow performance of different optical flow backbones. Therefore, the proposed common space adaptation is a plug-and-play training framework that can solve the nighttime optical flow problems.
>
> [1] Deng C, et al. Explicit motion disentangling for efficient optical flow estimation. ICCV, 2023.

---

> ### Author Response · Authors · 2023-11-15
>
> **Q3:** Application for low-light and high-speed scenarios.
>
> **A:** Low light and high speed are two of the more challenging scenarios for nighttime scenes. In terms of imaging mechanism, conventional frame camera records the absolute luminance with a fixed exposure time via global scan. In nighttime scenes, conventional frame camera would inevitably face a dilemma between the long exposure time of low-light scenarios and motion blur of high-speed scenarios. In contrast, the event camera reacts to changes in light intensity, rather than integrating photons during the exposure time of each frame [2]. Each pixel works independently and returns a signal only when an intensity change is detected. Compared with the conventional frame camera, the event camera can sense the dynamic changes with higher temporal resolution (microsecond) and higher dynamic range, thus compensating for frame camera in nighttime scenes, such as, nighttime image enhancement [3] and nighttime deblurring [4].
>
> Similarly, the event camera can also assist the frame camera to learn the motion patterns in nighttime low-light and high-speed scenarios. To discuss the impact of the event on the optical flow in nighttime low-light and high-speed scenes, as shown in the following table, we use the coaxial optical system (seeing Fig. 1 in the supplementary PDF) to collect the spatiotemporally-aligned image sequences and events stream with various illumination (e.g., 3.5 lux, 9.2 lux and 12.7 lux) and various driving speed (e.g., 50 km/h, 70 km/h and 80 km/h) to quantitatively compare the optical flow performance. As for the optical flow label, since it is difficult to directly obtain the dense optical flow labels, we manually mark 100 pairs of corresponding corner points for each two adjacent images, and calculate the relative displacement between the corner points as the sparse optical flow labels. In addition, we choose EPE as the evaluation metric.
>
> |                                              | Low light |         |          | High speed |         |         |
> | :------------------------------------------: | :-------: | :-----: | :------: | :--------: | :-----: | :-----: |
> |                    Method                    |  3.5 lux  | 9.2 lux | 12.7 lux |  50 km/h   | 70 km/h | 80 km/h |
> |              our flow baseline               |   4.26    |  3.98   |   3.67   |    3.65    |  4.53   |  5.74   |
> |       our flow baseline, w/ only event       |   2.05    |  2.01   |   1.63   |    1.75    |  1.75   |  1.80   |
> | our flow baseline, w/ event, w/ common space |   1.52    |  1.44   |   1.10   |    1.14    |  1.15   |  1.19   |
>
> We have two observations. First, the event camera can greatly improve optical flow performance in both nighttime low-light and nighttime high-speed scenes. In high-speed scenes, the proposed method is robust to various speeds, and the optical flow performance remains unchanged, demonstrating the advantage of high temporal resolution of the event camera. In low-light scenes, as the illumination becomes lower, the optical flow metric (EPE) trend becomes larger obviously. This shows that, although the event camera has the advantage of high dynamic range, too low illumination would also interfere with the optical flow performance. The main reason is that, under low light conditions, the event noise is intensified. The event noise and the valid signal event are both 0-1 pulses, and their difference is very small, which affects the optical flow. In the future, we will further consider the impact of noise and achieve optical flow estimation under extremely low-light scenes. Second, the proposed common space can further improve the upper limit of optical flow, indicating that common space can serve as a bridge to reinforce the feature alignment between event and nighttime image domains.
>
> [2] Cabriel C, et al. Event-based vision sensor for fast and dense single-molecule localization microscopy. Nature Photonics, 2023.
>
> [3] Liang J, et al. Coherent Event Guided Low-Light Video Enhancement. ICCV, 2023.
>
> [4] Qi Y, et al. E2NeRF: Event Enhanced Neural Radiance Fields from Blurry Images. ICCV, 2023.

---

> ### Author Response · Authors · 2023-11-19
>
> Dear Reviewer koka,
>
> Sincere gratitude for your comment. We have addressed all the comments raised by you. Item by item responses to the your comments are listed above this response. We are looking forward to your feedback.
>
> Best wishes.

---

> ### Author Response · Authors · 2023-11-21
>
> Dear Reviewer koka,
>
> We appreciate your thorough review. We have actively conducted various experiments to address your valuable questions. As the deadline for discussions is approaching, we look forward to your prompt feedback.
>
> Best wishes.

---

### Author Response · Authors · 2023-11-15

We first thank the reviewers for affirming our main contributions: **significant novelty** with common space adaptation to **any degraded scene understanding tasks** [koka, R9ad], **reasonable and complementary** appearance-boundary adaptation framework with **good generalization** [koka, hzUU, R9ad], well-conducted experiments with **state-of-the-art performance** [koka, hzUU], well-presented figures and tables [hzUU], clear writing [R9ad].

As for the main problems, the reviewer koka mainly focuses on the computational complexity and efficiency of the proposed framework, and the reviewer hzUU suggests a discussion on the limitation, and the reviewer R9ad cares about the complexity of the loss function.

In conclusion, we deeply thank the reviewers for their appreciation of our contributions, and for the valuable suggestions that advance the integrity of this work.

---

### Meta-Review · Area_Chair_sNtm · 2023-12-06

**Metareview:**

The final ratings are all positive with 1 strong acceptance. The paper solves the nighttime optical flow task via learning common latent spaces. The reviewers appreciate the technical contributions and comprehensive experimental results, in which the rebuttal addressed the raised questions well. Hence, the acceptance rating is recommended.

**Justification For Why Not Higher Score:**

While it is nice to show visual results on real data to demonstrate the generalization ability in Fig. 6 of the main paper, it would be more informative to show quantitative results, especially when the evaluated nighttime datasets are mostly paired with the daytime datasets for training. Also, it would be highly encouraged to release the implementation and benefit the community due to the high complexity of the proposed method.

**Justification For Why Not Lower Score:**

N/A

---

### Decision · Program_Chairs · 2024-01-16

Accept (spotlight)